# Maternal obesity during pregnancy disrupts iron homeostasis and promotes fetal hypoxia in the mouse

Adriana Córdova-Casanova[1] , Isabella Inzani[1] , Antonia Hufnagel[1], Dino A Giussani[2,3,4,5] , Denise S. Fernandez-Twinn[1,2,3] and Susan E Ozanne[1,2,3,4]

[1]University of Cambridge Institute of Metabolic Science and Medical Research Council Metabolic Diseases Unit, Addenbrookes Hospital, Cambridge, UK

[2]Loke Centre for Trophoblast Research, University of Cambridge, Cambridge, UK

[3]Cambridge Strategic Research Initiative in Reproduction, University of Cambridge, Cambridge, UK

[4]British Heart Foundation, Cambridge Cardiovascular Centre for Research Excellence, University of Cambridge, Cambridge, UK

[5]Department of Physiology, Development, and Neuroscience, University of Cambridge, Cambridge, UK

Handling Editors: Laura Bennet & Janna Morrison

Adriana Córdova-Casanova and Isabella Inzani contributed equally to this work.

The peer review history is available in the Supporting Information section of this article (https://doi.org/10.1113/JP288635#support-information-section).

**Abstract figure legend** Maternal obesity induces systemic inflammation and high insulin and leptin levels, potentially promoting hepcidin release into circulation and thus iron sequestration in specific tissue compartments. This results in

 

**Adriana Córdova Casanova** is a postdoctoral researcher in the laboratory of Professor Susan Ozanne at the Institute of Metabolic Science, University of Cambridge. She obtained her Master and PhD in Cellular and Molecular Biology from the Pontifical Catholic University of Chile, where she studied signalling mechanisms in skeletal muscle fibrosis under the supervision of Professor Enrique Brandan. Her current postdoctoral studies are focused on cellular communication in the context of maternal programming. **Isabella Inzani** completed her PhD in Clinical Biochemistry at the University of Cambridge investigating the developmental programming of offspring cardiometabolic disease by maternal obesity. Her research focused on exploring how the *in utero* environment can affect fetal development and the impact of this on long term offspring health.

D. S. Fernandez-Twinn and S. E. Ozanne contributed equally to this work and share senior authorship.

low iron availability in the mother with potential detrimental effects on the fetus, including hypoxia. Fetal hypoxia may be associated with short- and long-term negative outcomes for offspring cardiometabolic health. Created in BioRender.com (University of Cambridge Central Licence).

**Abstract** Research in both humans and animals has consistently demonstrated that diverse complications during pregnancy impact on the risk of chronic diseases in the offspring. In many settings, over half of women are either overweight or living with obesity during pregnancy. This has short- and long-term impacts on offspring health. The mechanisms mediating changes in the fetal environment that may trigger developmental origins of future cardiometabolic risk in the offspring are not fully elucidated. In this study, using an established mouse model, we aimed to determine whether obesity during pregnancy causes fetal hypoxia and to explore potential underlying mechanisms. We showed that fetal hypoxia is a key component of the *in utero* obesogenic environment at E13.5/0.7 of gestation. Concomitantly, obese dams exhibit low iron levels, as well as higher circulating levels of hepcidin and C-reactive protein. We also showed that placental structure and efficiency are not affected by maternal obesity at E13.5, suggesting that the reduction in oxygen delivery to the fetus was not a consequence of placental dysfunction at this stage of gestation. We conclude that maternal obesity-induced iron deficiency and fetal hypoxia are important mechanisms by which obesity during pregnancy impacts offspring health. Furthermore, iron deficiency in mothers with obesity is a tractable therapeutic target for intervention that could prevent transmission of poor cardiometabolic health from mother to child.

(Received 3 February 2025; accepted after revision 27 October 2025; first published online 23 November 2025)

**Corresponding author** A. Córdova-Casanova: University of Cambridge Institute of Metabolic Science and Medical Research Council Metabolic Diseases Unit, Addenbrookes Hospital, Cambridge, UK. Email: ap2427@cam.ac.uk

**Key points**

- Diet-induced maternal obesity resulted in fetal, but not placental, hypoxia.
- Pregnant mice with obesity had lower circulating iron levels, along with dysregulation of key molecules involved in iron homeostasis, such as transferrin and the hormone hepcidin.
- Body weight, fat mass, circulating insulin and hepcidin levels in mothers with obesity were significantly correlated with the degree of fetal hypoxia, suggesting they were interrelated.

## Introduction

Obesity is defined as abnormal adipose tissue accumulation in the body and has been strongly associated with the development of chronic metabolic diseases, representing one of the major concerns for human health. By 2030, it is projected that approximately 3 billion adults will be affected by overweight and obesity (World Obesity Federation, 2025). This prevalence extends to pregnant populations, with nearly half of women affected by overweight (body mass index (BMI) $\geq 25$ kg/m²) or obesity (BMI $\geq 30$ kg/m²) at the onset of pregnancy (Hill et al., 2019; Wang et al., 2019). Obesity during pregnancy is associated with an increased risk of complications, including preeclampsia, gestational diabetes mellitus (GDM) (Fitzsimons et al., 2009; Stephenson et al., 2018) and an increased risk of poor cardiometabolic health in the offspring (Gaillard et al., 2014; Perng et al., 2014). Pre-clinical studies in experimental animal models support

the observations in humans and show a causal relationship between obesity during pregnancy and offspring cardiometabolic disorders that occur independently of genotype (reviewed in Cochrane et al., 2024). For example, using a mouse model of maternal diet-induced obesity, we and others have shown that maternal obesity leads to offspring insulin resistance (Fernandez-Twinn et al., 2014), fatty liver (Alfaradhi et al., 2014; Oben et al., 2010), hyperphagia-induced obesity (Dearden et al., 2024) and cardiovascular dysfunction during adulthood (Beeson et al., 2018; Samuelsson et al., 2008; Vaughan et al., 2022). However, the component(s) of the obesogenic in utero environment that mediate such adverse effects on the offspring are not understood.

Chronic fetal hypoxia is a common complication during pregnancy, which has established long-term adverse effects on the structure and function of tissues and organs in offspring, leading to an increased risk of poor cardiometabolic health (Giussani et al., 2014; Giussani,

2021; Patterson & Zhang, 2010). Fetal hypoxia can occur as a result of reduced uterine or umbilical blood flow, high placental vascular resistance, high-altitude pregnancy or maternal anaemia (Hutter et al., 2010). Maternal anaemia usually results from an inability to meet the increased iron requirement to support maternal and fetal cellular processes with advancing gestation (Bothwell, 2000). Iron deficiency in pregnancy is known to induce cardiac proteome changes with deregulation of mitochondrial proteins in a rat model (Holody et al., 2025), to affect cardiovascular development in mouse offspring (Kalisch-Smith et al., 2021) and to be a risk factor for congenital heart disease in humans (Chou et al., 2016). One cause of reduced iron availability is chronic systemic inflammation (Ganz, 2019; Opasich et al., 2005; Weiss et al., 2019). During chronic inflammation, elevated hepcidin leads to sequestration of iron within hepatocytes, enterocytes and macrophages, reducing its availability for haemoglobin synthesis, thereby hindering red blood cell production. Pregnancy is a pro-inflammatory state per se, and obesity exacerbates this condition (Bernhardt et al., 2022; Ellulu et al., 2017; Khanna et al., 2022). Therefore, pregnant women affected by obesity could be at greater risk of developing iron deficiency (Garcia-Valdes et al., 2015; Wawer et al., 2021), increasing the risk of fetal hypoxia. However, this has not been investigated.

Therefore, the aims of the current study were to: (1) establish if obesity during pregnancy causes fetal hypoxia and (2) explore changes in inflammatory pathways and iron homeostasis as a feasible mechanism to explain the origins of adverse cardiovascular effects in offspring of mothers affected by obesity during pregnancy.

## Methods

### Ethical approval

Animal studies were performed following review and approval by the University of Cambridge Animal Welfare and Ethical Review Body and in accordance with the UK Animals (Scientific Procedures) Act 1986, under the animal project licence PP8498895 issued by the UK Home Office. Experiments were designed and reported with reference to the ARRIVE guidelines (Kilkenny et al., 2010). The experiments comply with the policies and regulations of *The Journal of Physiology* (Grundy, 2015). In total, 18 mouse pregnancies (nine control and nine obese) were used in this study. All animal studies were performed at the University of Cambridge (UK).

### Diet-induced obesity mouse model

Our study utilised a well-established diet-induced mouse model of maternal obesity (Fernandez-Twinn et al.,

2017). Briefly, female C57BL/6J mice (Charles River laboratories, Harlow, UK) were randomly assigned to either a control chow diet (CCD; SAFE A05), or an obesogenic high fat diet (HFD; SAFE U8954A01R 00279; Safe Diets, Augy, France), containing 45% total calories from fat, fortified with mineral mix AIN93G (Special Diet Services, Witham, UK) and supplemented with condensed milk composed of 55% sugar and 8% fat (Nestle, Crawley, UK). The iron composition of the CCD and HFD diets was 270 mg of iron/kg and 67 mg of iron/kg, respectively. We calculate that daily iron intake corresponds to at least 1.62 mg and 0.268 mg for CCD and HFD, respectively. At approximately 6 weeks of age, and 2 weeks after starting on their respective diets, females were mated with CCD-fed males for their first pregnancy. Dams remained on their respective diets during pregnancy and lactation. At the age of 3–4 months, both control lean and obese primiparous females were mated for a second time with CCD-fed males. For assessment of placental and fetal hypoxia at E13.5/0.7 of gestation, 90 min before being killed by exposure to increasing concentrations of $CO_2$, dams were injected intraperitoneally with 60 mg/kg pimonidazole, which binds thiol groups in proteins, forming pimonidazole–protein adducts in hypoxic tissues. Fetuses were sexed by PCR using genomic DNA isolated from tail biopsies as previously described (Hufnagel et al., 2022).

### Time-domain nuclear magnetic resonance imaging

To determine body fat mass composition, time-domain nuclear magnetic resonance imaging (Minispec Plus, Bruker, Billerica, MA, USA) was performed. Measurements were taken in dams pre-mating and at weekly intervals during pregnancy at day E0.5 (day of plug), E7.5 (=0.4 of gestation), and E13.5 (=0.7 of gestation as term is at 19–21 days in the mouse). For all measurements, the animals were alive, fed, and non-anaesthetised. The measurements were always performed before 09.00 h.

### Blood and serum measurements

Tail blood glucose levels were analysed using the AlphaTRAK2 system (Zoetis, Parsippany, NJ, USA). Dam fed serum was used in immunoassays to measure levels of insulin (Ultra-Sensitive Mouse Insulin ELISA Kit, CrystalChem, Elk Grove Village, IL, USA), leptin (Mouse Leptin ELISA Kit, CrystalChem), adiponectin (Mouse Adiponectin ELISA Kit, CrystalChem), ferritin (Mouse Ferritin ELISA Kit, Abcam, Cambridge, UK), transferrin (Mouse Transferrin ELISA Kit, Abcam) and hepcidin (Mouse hepcidin ELISA kit, Elabscience, Houston, Texas, USA) according to the manufacturer's instructions. Iron

(DF85 test,Dimension EXL analyser, Siemens, Forcheim, Germany), a cytokine panel (10-plex proinflammatory panel 1 mouse kit, Meso Scale Discovery, RockVille, MD, USA), and C-reactive protein (CRP; antibodies from R&D, US/DELFIA platform from PerkinElmer, Shelton, CT, USA) were measured by the IMS-MRL Core Biochemical Assay Laboratory. For haematocrit measurement, the head was removed from E18.5 fetuses to obtain blood in capillary tubes (Hirschmann-Laborgeräte, Eberstadt, Germany), which were centrifuged at 10,000 *g*, and the haematocrit was determined using a capillary tube reader. Maternal haematology profile was analysed using 20 µl of EDTA-anticoagulated maternal blood obtained from a tail nick and processed according to instructions on the Element HT5 Auto Haematology Analyser (Antech diagnostics, fountain valley, CA, USA.)

### Histological analysis

To control for within-litter differences, one male and one female fetus from each litter were used for histological analyses. Mouse E13.5 fixed whole fetuses and placentas were processed, embedded in paraffin, and sectioned using a microtome (Leica Microsystems, Wetzlar, Germany) at 3 µm thickness. Fetuses were sectioned in the frontal plane at mid-cardiac level, and placentas at the level of the mid-placenta. Whole fetal and placental sections were stained using an antibody against pimonidazole (hybridoma clone 4.3.11.3, Hypoxyprobe, Burlington, MA, USA; 1.2 µg/ml diluted in antibody diluent; Vector Laboratories, Burlingame, CA, USA) and the biotinylated anti-mouse secondary antibody (HRP/DAB (ABC) Detection IHC Kit; Abcam). Sections were counter-stained with haematoxylin QS (Vector Laboratories) and coverslip-mounted using DPX (Sigma-Aldrich, St Louis, MO, USA). In addition, Alizarin Red S staining was performed in adjacent sections to assess placental calcification. Sections were stained in Alizarin Red S solution for 5 min (1% aqueous solution pH 6.4, ammonium hydroxide) and then counterstained with Fast Green (0.05% Fast Green solution (Sigma-Aldrich) in 0.2% acetic acid). Sections were imaged using a Slide Scanner AxioScan Z1 microscope (Carl Zeiss Microscopy, Jena, Germany). Fetal and placental tissue regions were identified based on morphology, manually annotated, and then analysed with QuPath (v0.3.0370, open-source software) to assess pimonidazole stain intensity using HALO (v3.2, Indica Labs, Albuquerque, NM, USA). Placental vascularization was assessed by immuno-detection of CD31, a marker of endothelial cells.

### Statistical methods

All statistical analyses were performed using Prism10 software (GraphPad Software, Boston, MA, USA). Unpaired two-tailed Student's *t* tests were used when only two groups were compared (control *versus* obese). Two-way analysis of variance (ANOVA) was performed to estimate the effect of two independent variables (maternal diet and fetal sex) followed by Tukey' *post hoc* testing to isolate differences between groups. Pearson's coefficient was determined for correlation analyses. Statistically significant was considered for *P*-values as follows: $^*P \leq 0.05$; $^{**}P \leq 0.01$; $^{***}P \leq 0.001$ and $^{****}P \leq 0.0001$. Data are presented as means $\pm$ SD.

## Results

### Maternal obesity model

As expected, in our model of maternal diet-induced obesity, obese dams had increased maternal body weight (Fig. 1*A*), maternal fat mass (Fig. 1*B*) and higher gestational weight gain (Fig. 1*C*) from day of plug to E13.5/0.7 of gestation, compared with control dams. At E13.5 fed dam blood glucose (Fig. 1*D*), serum insulin (Fig. 1*E*) and serum leptin (Fig. 1*F*), but not serum adiponectin (Fig. 1*G*), were significantly higher in obese dams compared with controls. Organs collected *post mortem* at E13.5 showed that obese dams had significantly increased liver, left kidney, and gonadal, intraperitoneal and retroperitoneal fat pad weights compared with controls (Table 1). When expressed relative to body weight, obese dams had persistently increased gonadal, intraperitoneal and retroperitoneal fat pad weights compared with control dams (Table 1). Together, these data indicate that our model resembles human obesity parameters (Busebee et al., 2023).

### Maternal obesity resulted in fetal tissue hypoxia at E13.5

Fetal hypoxia at E13.5 was assessed by measuring the intensity of pimonidazole staining (Fig. 2*A*), a widely used indicator of tissular hypoxia. Our results showed that fetal tissues from obese pregnancies were more hypoxic than those from control pregnancies when assessed over-all, in the whole fetal torso (Fig. 2*B*), and when assessed compartmentally in the heart (Fig. 2*C*), liver (Fig. 2*D*) and brain (Fig. 2*E*) in a non-sex-dependent manner. The degree of hypoxia was similar in all studied tissues (Fig. 2*B–E*). To elucidate whether our model showed a brain-sparing effect during chronic fetal hypoxia (a prioritisation of the supply of oxygen to the fetal brain over other tissues; Giussani, 2016), we calculated the ratio of pimonidazole staining intensity between the fetal brain and whole fetal torso. No statistical differences were found between groups (Fig. 2*F*).

As an initial exploration to determine which maternal factors may be driving fetal hypoxia in obese pregnancy,

correlations were calculated between indices of fetal hypoxia (anti-pimonidazole staining intensity) and dam body weight, dam fat mass and circulating levels of insulin, leptin, glucose and adiponectin (Fig. 3; Table 2). Fetal hypoxia positively correlated with dam E13.5 body weight (Fig. 3*A*), fat mass (Fig. 3*B*) and serum insulin levels (Fig. 3*C*). However, no correlations between fetal hypoxia and dam E13.5/0.7 of gestation serum leptin (Fig. 3*D*), blood glucose (Fig. 3*E*) or serum adiponectin (Fig. 3*F*) were observed. These data confirmed that the degree of fetal hypoxia is associated with increasing maternal adiposity and hyperinsulinaemia.

### Maternal obesity did not modify placental efficiency, hypoxia, calcifications, or vasculature changes in the E13.5 placenta

To determine whether fetal hypoxia was due to adverse effects in the placenta at E13.5/0.7 of gestation, we evaluated various placental parameters, including efficiency, hypoxia, calcification and vascularisation in the placenta at this stage of gestation. Maternal obesity did not affect E13.5 fetal or placental weight (Fig. 4*A* and *B*). No differences in placental efficiency, calculated as fetal body weight divided by placental weight, were detected at this stage of gestation (Fig. 4*C*). Placental hypoxia was also assessed by measuring the intensity of anti-pimonidazole staining across the tissue. Our results showed no differences in the degree of hypoxia in the whole placenta (Fig. 4*D* and *E*), the labyrinthine zone, the junctional zone, or the decidua (Fig. 4*F*, *G* and *H*) between placentas from control and obese dams. There were no correlations between any of the measured maternal obesity-related factors (body weight, fat mass, glucose, insulin, leptin or adiponectin) and placental hypoxia (Table 2). Alizarin Red S was used to stain calcium deposits in the placenta. No differences were observed in calcification between control and obese groups in the E13.5/0.7 of gestation placentas, and the percentage staining was very low in both groups at this stage of pregnancy (Fig. 4*I* and *J*). To study vascularisation

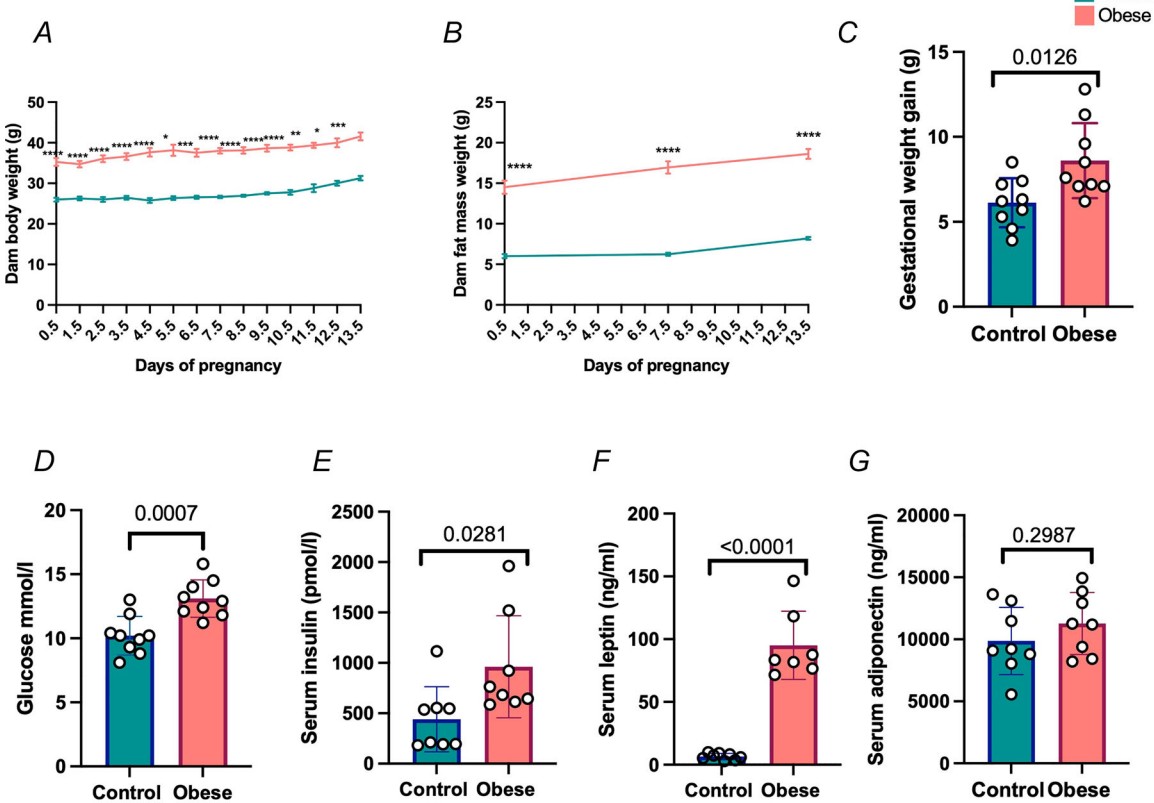

**Figure 1. Maternal high fat and high sugar diet feeding resulted in an obese dam phenotype**
*A*, daily maternal body weight from day of plug until mid-gestation (E13.5/0.7 of gestation). *B*, weekly maternal fat mass measured by time-domain nuclear magnetic resonance imaging from day of plug to E13.5. *C*, gestational dam weight gain between day of plug and E13.5. *D*, dam E13.5 blood glucose levels. *E*, dam E13.5 serum insulin levels. *F*, dam E13.5 serum leptin levels. *G*, dam E13.5 serum adiponectin levels. All data are presented as means ± SD and statistically assessed by fitting a mixed effects model (*A–B*; ****$P < 0.0001$; ***$P < 0.001$; **$P < 0.01$; *$P < 0.05$) or Student's unpaired *t* test (*C–G*; *P*-values indicated). *n* = 7–9 per condition.

**Table 1. Additional dam pregnancy and post-mortem data**

| | Control | Obese | P |
|---|---|---|---|
| Dam | | | |
| Pre-pregnancy lean mass (g) | 14.86 ± 0.89 | 14.17 ± 1.19 | 0.1873 |
| E0.5 lean mass (g) | 15.12 ± 1.48 | 13.96 ± 0.95 | 0.0651 |
| E6.5 lean mass (g) | 15.94 ± 0.71 | 14.06 ± 1.08 | 0.0006 |
| E13.5 lean mass (g) | 17.04 ± 1.27 | 15.59 ± 1.18 | 0.0244 |
| Pre-pregnancy free fluid mass (g) | 1.59 ± 0.30 | 2.38 ± 0.49 | 0.0068 |
| E0.5 free fluid mass (g) | 1.87 ± 0.18 | 2.49 ± 0.41 | 0.0012 |
| E6.5 free fluid mass (g) | 1.62 ± 0.14 | 2.40 ± 0.36 | <0.0001 |
| E13.5 free fluid mass (g) | 3.48 ± 0.33 | 5.07 ± 0.49 | <0.0001 |
| Dam – absolute organ weights | | | |
| Heart weight (g) | 0.18 ± 0.02 | 0.19 ± 0.02 | 0.2001 |
| Liver weight (g) | 1.87 ± 0.28 | 2.26 ± 0.33 | 0.0161 |
| Left kidney weight (g) | 0.18 ± 0.02 | 0.19 ± 0.01 | 0.0328 |
| Right kidney weight (g) | 0.19 ± 0.03 | 0.21 ± 0.02 | 0.0678 |
| Gonadal fat pad weight (g) | 0.37 ± 0.10 | 2.39 ± 0.59 | <0.0001 |
| Intraperitoneal fat weight (g) | 0.24 ± 0.03 | 0.93 ± 0.18 | <0.0001 |
| Retroperitoneal fat pad weight (g) | 0.20 ± 0.03 | 1.15 ± 0.48 | <0.0001 |
| Brain weight (g) | 0.51 ± 0.03 | 0.49 ± 0.02 | 0.0849 |
| Dam – relative to dam E13.5 lean mass | | | |
| Relative heart weight (g/g) | 0.010 ± 0.00 | 0.012 ± 0.00 | 0.0244 |
| Relative liver weight (g/g) | 0.111 ± 0.02 | 0.144 ± 0.01 | 0.0020 |
| Relative left kidney weight (g/g) | 0.010 ± 0.00 | 0.012 ± 0.00 | <0.0001 |
| Relative right kidney weight (g/g) | 0.010 ± 0.00 | 0.014 ± 0.00 | <0.0001 |
| Relative gonadal fat pad weight (g/g) | 0.021 ± 0.01 | 0.154 ± 0.04 | <0.0001 |
| Relative intraperitoneal fat weight (g/g) | 0.014 ± 0.00 | 0.059 ± 0.01 | <0.0001 |
| Relative retroperitoneal fat pad weight (g/g) | 0.011 ± 0.00 | 0.073 ± 0.03 | <0.0001 |
| Relative brain weight (g/g) | 0.029 ± 0.00 | 0.031 ± 0.00 | 0.1916 |
| Dam – relative to dam body weight | | | |
| Relative heart weight (g/g) | 0.54 ± 0.04 | 0.43 ± 0.04 | <0.0001 |
| Relative liver weight (g/g) | 5.79 ± 1.07 | 5.19 ± 0.60 | 0.1647 |
| Relative left kidney weight (g/g) | 0.54 ± 0.04 | 0.45 ± 0.03 | <0.0001 |
| Relative right kidney weight (g/g) | 0.57 ± 0.06 | 0.49 ± 0.04 | 0.0017 |
| Relative gonadal fat pad weight (g/g) | 1.12 ± 0.30 | 5.49 ± 1.26 | <0.0001 |
| Relative intraperitoneal fat weight (g/g) | 0.73 ± 0.11 | 2.15 ± 0.32 | <0.0001 |
| Relative retroperitoneal fat pad weight (g/g) | 0.62 ± 0.08 | 2.64 ± 1.06 | <0.0001 |
| Relative brain weight (g/g) | 1.56 ± 0.06 | 1.12 ± 0.07 | 0.084 |
| Litter | | | |
| Gravid uterus weight (g) | 3.37 ± 0.85 | 3.96 ± 0.62 | 0.2131 |
| Litter size (n) | 7.78 ± 1.72 | 9.00 ± 1.23 | 0.1012 |
| Sex ratio (% male) | 53.59 ± 17.70 | 52.53 ± 15.80 | 0.8942 |

Differences between control and obese groups were tested by using Student's unpaired $t$ test or non-parametric equivalent as appropriate. All values are shown as means ± SD.

of the placentas, the percentage of area stained for CD31, a marker of endothelial cells, was analysed. No significant differences were observed between placentas from control and obese dams (Fig. 4K and L).

## Maternal iron homeostasis was disrupted by obesity

To establish whether iron handling could be involved in promoting the hypoxic phenotype in fetuses of obese mothers, the levels of iron in the dam serum were analysed. There was a statistically significant reduction in serum iron levels in obese dams compared with controls (Fig. 5A), although this did not result in anaemia, as assessed by haematocrit and haemoglobin levels (Fig. 5B and C). Ferritin, an indicator of iron stores in the body, and transferrin, the main transporter for circulating iron (Bouri & Martin, 2018), were analysed. There were no differences in ferritin levels (Fig. 5D), but higher transferrin levels (Fig. 5E) were observed in serum from obese dams compared with controls.

As obesity is a recognised low degree chronic inflammatory state, and inflammation is known to activate hepcidin production and secretion (Wang & Babitt, 2016), we explored the serum levels of 8 inflammation-related markers and found an upward trend in some pro-inflammatory markers (Table 3), with a statistically significant increase in C-reactive protein (CRP) in obese dams (Fig. 5*F*). We also detected an increase in the levels of white blood cells (Fig. 5*G*). Consistent with these findings, maternal serum hepcidin levels were significantly raised in obese mothers (Fig. 5*H*). To understand whether the hepcidin levels were related to fetal hypoxia, the correlation between circulating maternal hepcidin levels and the degree of hypoxia observed in E13.5 fetuses was assessed. Our results showed a statistically significant positive correlation

between these two variables (Fig. 5*I*). Moreover, maternal circulating hepcidin levels showed a strong positive correlation with obesity-related factors in dams, including maternal body weight, maternal fat mass, maternal leptin and maternal insulin at E13.5 (Table 4). We performed these correlation tests separately in both groups (control and obese), finding a statistical difference when assessing maternal hepcidin levels and body weight just in the obese group (Fig. 5*J*). Additionally, hepcidin levels were significantly correlated with key iron metabolism parameters, including serum iron, ferritin and transferrin levels in dams (Table 4). As measuring the fetal iron levels at E13.5 was not possible for technical reasons, we tested the fetal haematocrit at E18.5 as an indirect indicator of iron levels. We observed a statistically significant reduction in the offspring of obese dams (Fig. 5*K*).

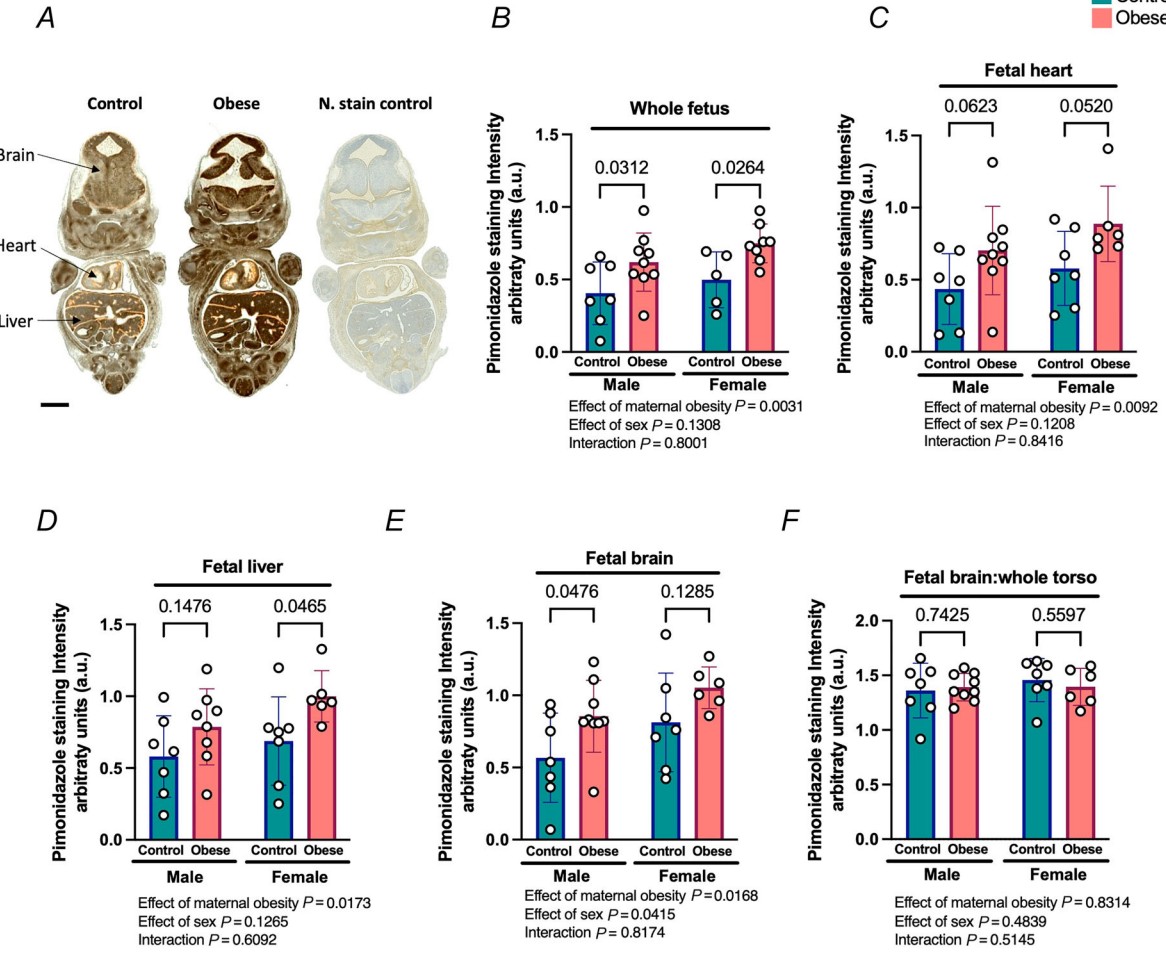

**Figure 2. Maternal obesity resulted in fetal hypoxia at E13.5 without brain sparing**
*A*, representative images of control and obese fetuses stained with anti-pimonidazole antibody, and a negative stain control. Scale bars = 1 mm. *B–E*, intensity of staining for the tissue hypoxia marker pimonidazole in the whole fetus torso (*B*), fetal heart (*C*), fetal liver (*D*), and fetal brain (*E*). *F*, ratio of pimonidazole staining intensity of the fetal brain divided by the whole fetal torso. All data are presented as means ± SD and statistically assessed by two-way ANOVA. *P* values are indicated. *n* = 5–9 per condition.

## Discussion

Maternal obesity during pregnancy is a risk factor for adverse outcomes in the offspring in both the short- and the longer-term (Dearden & Ozanne, 2023; Stephenson et al., 2018). Research on experimental animal models (murine and non-human primates) of maternal obesity during pregnancy has consistently shown poor cardiovascular outcomes in the adult offspring (Beeson et al., 2018; Bertossa et al., 2024; Blackmore et al., 2014; Loche et al., 2018; Samuelsson et al., 2008; Vaughan et al., 2022). However, the mechanisms linking what the fetus experiences *in utero* in an obesogenic environment to the programmed increase in cardiovascular risk are not well understood (Cochrane et al., 2024). In the current study, we establish that inflammation and disruption of iron homeostasis in obese pregnancies are linked to fetal hypoxia. At E13.5/0.7 of gestation, we observed consistently higher levels of hypoxia in the heart, liver and brain in fetuses of obese pregnancies compared with controls. Conversely, placental factors, such as efficiency, hypoxia, calcification and vasculature, were not affected by maternal obesity at this gestational age. This indicates that the placental abnormalities we previously observed at E18.5/0.9 of gestation, such as

obesity-induced placental calcification and resultant lower placental efficiency (Hufnagel et al., 2022), emerge later in pregnancy. Therefore, the development of fetal hypoxia at E13.5/0.7 of gestation precedes and is thereby triggered by mechanisms independent of the onset of adverse placental factors observed at 0.9 of gestation in this mouse model of maternal obesity during pregnancy. The lack of a fetal brain-sparing effect triggered by fetal hypoxia at E13.5/0.7 of gestation is likely due to immaturity of the chemo-reflex, endocrine and local redox mechanisms that initiate and maintain this fetal cardiovascular defence (Giussani, 2016). It is known that the fetal defence to hypoxia matures with advancing gestational age, in parallel with the maturation of the regulation of fetal cardiovascular function (Fletcher et al., 2006; Giussani, 2016; Jellyman et al., 2020), and that the fetal peripheral vasoconstriction helping to redistribute blood flow towards the fetal brain does not occur at 0.6−0.7 of gestation in other species such as the sheep (Iwamoto et al., 1989).

During pregnancy, iron requirements increase gradually throughout gestation (Bothwell, 2000). Thus, iron supplementation is a frequent therapy to correct iron deficiency in pregnant women (Zhao et al., 2015) to prevent negative outcomes, including fetal growth restriction (Kozuki et al., 2012). Prenatal iron deficiency induced by

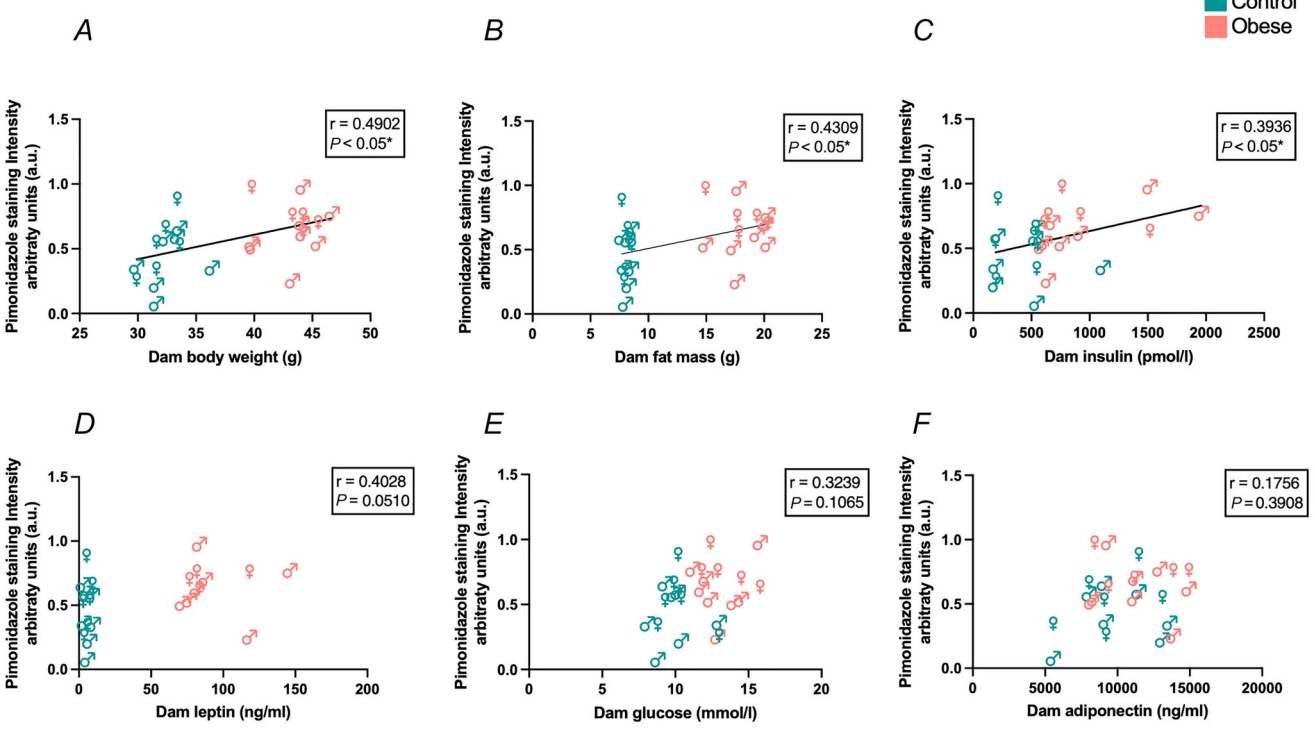

**Figure 3. Fetal hypoxia correlated with dam factors**
Correlations between fetal whole torso anti-pimonidazole staining intensity and dam E13.5. *A*, body weight; *B*, fat mass; *C*, serum insulin; *D*, serum leptin; *E*, blood glucose; and *F*, serum adiponectin. Blue symbols: control dams/litters; red symbols: obese dams/litters; ♂: male fetuses; ♀: female fetuses. All correlations were assessed by calculation of the Pearson's correlation coefficient. *r* and *P*-values are indicated. *n* = 5–8 per condition.

**Table 2. Correlations between dam factors and individual fetal and placental zones**

|  | r | P |
|---|---|---|
| **Fetal hypoxia correlations** | | |
| Fetal heart *vs.* dam fat mass | 0.40 | 0.0425 |
| Fetal heart *vs.* dam body weight | 0.45 | 0.0216 |
| Fetal heart *vs.* dam glucose | 0.36 | 0.0637 |
| Fetal heart *vs.* dam insulin | 0.04 | 0.0571 |
| Fetal heart *vs.* dam leptin | 0.37 | 0.0754 |
| Fetal heart *vs.* dam adiponectin | 0.05 | 0.8275 |
| Fetal liver *vs.* dam fat mass | 0.37 | 0.0695 |
| Fetal liver *vs.* dam body weight | 0.43 | 0.0324 |
| Fetal liver *vs.* dam glucose | 0.32 | 0.1163 |
| Fetal liver *vs.* dam insulin | 0.37 | 0.0720 |
| Fetal liver *vs.* dam leptin | 0.35 | 0.1074 |
| Fetal liver *vs.* dam adiponectin | 0.17 | 0.4144 |
| Fetal brain *vs.* dam fat mass | 0.44 | 0.0241 |
| Fetal brain *vs.* dam body weight | 0.50 | 0.0088 |
| Fetal brain *vs.* dam glucose | 0.33 | 0.1010 |
| Fetal brain *vs.* dam insulin | 0.39 | 0.0510 |
| Fetal brain *vs.* dam leptin | 0.38 | 0.0678 |
| Fetal brain *vs.* dam adiponectin | 0.20 | 0.3239 |
| **Placental hypoxia correlations** | | |
| Whole placenta *vs.* dam fat mass | −0.07 | 0.7014 |
| Whole placenta *vs.* dam body weight | 0.01 | 0.9645 |
| Whole placenta *vs.* dam glucose | −0.10 | 0.6133 |
| Whole placenta *vs.* dam insulin | 0.20 | 0.3183 |
| Whole placenta *vs.* dam leptin | −0.02 | 0.9389 |
| Whole placenta *vs.* dam adiponectin | −0.01 | 0.9738 |
| Decidua *vs.* dam fat mass | −0.17 | 0.4184 |
| Decidua *vs.* dam body weight | −0.07 | 0.7416 |
| Decidua vs dam glucose | −0.22 | 0.2942 |
| Decidua *vs.* dam insulin | 0.06 | 0.7780 |
| Decidua *vs.* dam leptin | −0.14 | 0.5187 |
| Decidua *vs.* dam adiponectin | −0.02 | 0.9352 |
| Junctional zone *vs.* dam fat mass | −0.04 | 0.8457 |
| Junctional zone *vs.* dam body weight | 0.06 | 0.7871 |
| Junctional zone *vs.* dam glucose | −0.09 | 0.6553 |
| Junctional zone *vs.* dam insulin | 0.25 | 0.2221 |
| Junctional zone *vs.* dam leptin | 0.003 | 0.9899 |
| Junctional zone *vs.* dam adiponectin | 0.09 | 0.6510 |
| Labyrinthine zone *vs.* dam fat mass | 0.16 | 0.4100 |
| Labyrinthine zone *vs.* dam body weight | 0.18 | 0.3398 |
| Labyrinthine zone *vs.* dam glucose | −0.79 | 0.7014 |
| Labyrinthine zone *vs.* dam insulin | 0.33 | 0.0753 |
| Labyrinthine zone *vs.* dam leptin | 0.23 | 0.2368 |
| Labyrinthine zone *vs.* dam adiponectin | −0.06 | 0.7803 |

Correlations between intensity of pimonidazole staining in fetal and placental sections with dam E13.5 fat mass (g), body weight (g), fed blood glucose (mmol/l), serum insulin (pmol/l), serum leptin (mg/ml) and serum adiponectin (ng/ml). All correlations were assessed by calculation of the Pearson's correlation coefficient. r and P-values are indicated.

a low iron diet (3 mg iron/kg, representing 8.1% of the iron in the control diet) induces significant changes in mitochondrial function and content in the kidney and liver in rats in a sex dependent manner compared with the control group receiving 37 mg/kg of diet (Woodman et al., 2018). Previous reports in C57BL/6 pregnant mice showed that a diet containing at least 45 mg iron/kg did not affect the haemoglobin levels or the red blood cell volume and represents an adequate iron supply to support normal fetal growth (Hubbard et al., 2013). The obese

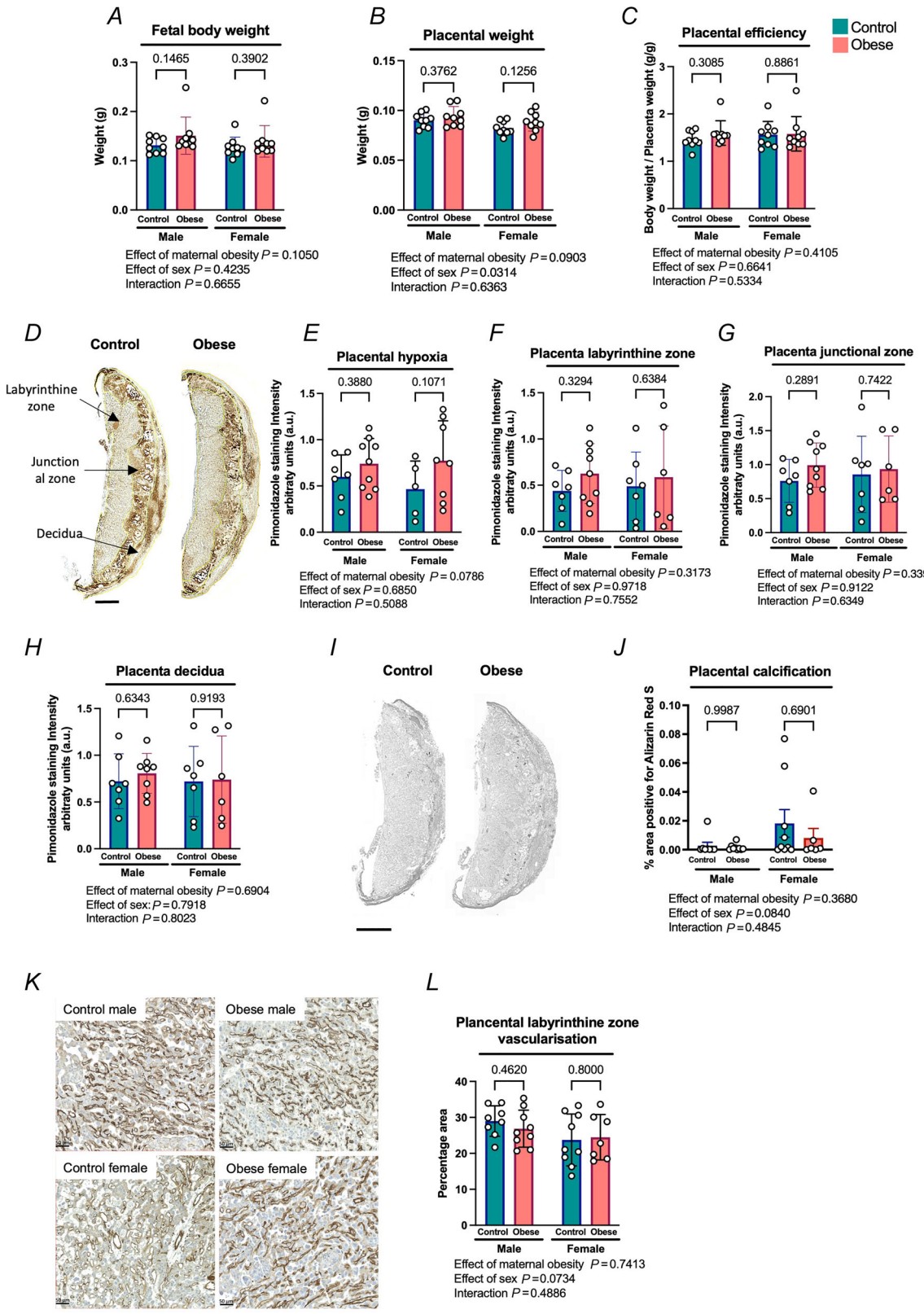

**Figure 4. Placental weight, vascularisation, calcification or intensity of staining for the tissue hypoxia marker pimonidazole were not affected by maternal obesity at E13.5**

*A*, fetal body weight. *B*, placenta weight. *C*, placental efficiency calculated as the ratio of fetal body weight to placental weight. *D*, representative images of control and obese placentas stained with anti-pimonidazole antibody.

Scale bar = 1 mm. *E–G*, intensity of pimonidazole stain in whole placenta (E) labyrinthine zone (*F*), placenta junctional zone (G), and placenta decidua (H). *I*, representative images of control and obese placentas stained with Alizarin Red S. *J*, percentage area of placental calcification by positive staining with Alizarin Red S. *K*, representative images of CD31-stained placentas showing vascular endothelial marker CD31 in the labyrinthine zone. Scale bar = 50 μm. *L*, percentage positive area stained for the CD31 in the placental labyrinthine zone. All areas were measured in mid-placental sections. All data are presented as means ± SD and statistically assessed by two-way ANOVA. *P*-values are indicated. *n* = 5–10 per condition.

**Table 3. Inflammatory panel markers in dam serum**

| Inflammatory marker | Control dams | Obese dams | *P* |
|---|---|---|---|
| IFN-$\gamma$ (pg/ml) | 0.53 ± 0.08 | 0.41± 0.09 | 0.0320 |
| CRP (mg/l) | 6.36 ± 1.20 | 9.37 ± 2.62 | 0.0140 |
| IL-2 (pg/ml) | 0.98 ± 0.20 | 1.26 ± 0.33 | 0.1007 |
| IL-5 (pg/ml) | 9.37 ± 5.02 | 6.28 ± 1.92 | 0.1095 |
| IL-6 (pg/ml) | 52.73 ± 26.17 | 492.40 ± 635.30 | 0.1182 |
| IL-10 (pg/ml) | 24.06 ± 11.49 | 19.93 ± 4.396 | 0.3618 |
| KC/GRO (pg/ml) | 63.33 ± 6.212 | 133.2 ± 107.4 | 0.1412 |
| TNF-$\alpha$ (pg/ml) | 12.53 ± 1.632 | 12.83 ± 1.831 | 0.7630 |

All values are shown as means ± SD. Significant differences between control and obese groups tested by Student's unpaired *t*-test. CRP, C-reactive protein; IFN-$\gamma$, interferon-gamma; IL-2, interleukin-2; IL-5, interleukin-5; IL-6, interleukin-6; IL-10, interleukin-10; KC/GRO, growth-regulated alpha protein; TNF-$\alpha$: tumour necrosis factor alpha.

**Table 4. Statistically significant correlations in the inflammation–iron metabolism–fetal hypoxia axis in the murine maternal obesity model**

| Correlation | *r* | *P* |
|---|---|---|
| Dam fat mass *vs*. IL6 | 0.5652 | 0.0026 |
| Dam fat mass *vs*. CRP | 0.5527 | 0.0023 |
| Dam fat mass *vs*. dam hepcidin | 0.8461 | <0.0001 |
| Dam fat mass *vs*. dam ferritin | −0.4835 | 0.0051 |
| Dam fat mass *vs*. fetal hypoxia | 0.3894 | 0.0493 |
| Dam hepcidin *vs*. dam iron | −0.4563 | 0.0376 |
| Dam body weight *vs*. hepcidin | 0.8799 | <0.0001 |
| Dam hepcidin *vs*. dam ferritin | −0.4706 | 0.0087 |
| Dam hepcidin *vs*. dam transferrin | 0.7181 | <0.0001 |
| Dam hepcidin *vs*. dam leptin | 0.8114 | 0.0004 |
| Dam hepcidin *vs*. dam insulin | 0.6297 | 0.0119 |
| Dam ferritin *vs*. fetal hypoxia | −0.3949 | 0.0459 |
| Dam iron *vs*. dam transferrin | −0.6258 | 0.0014 |
| Dam transferrin *vs*. CRP | 0.7222 | <0.0001 |

Correlations between different dam factors and fetal hypoxia. All correlations were assessed by calculation of Pearson's correlation coefficient. *r* and *P*-values are indicated.

dams in our study received an HFD diet containing 67 mg iron/kg, suggesting that the maternal iron deficiency observed in the obese relative to control dams was not caused by a nutritional iron deficiency.

Additional data in this study showed that obese dams had increased levels of circulating leucocytes and the inflammatory marker, CRP and hepcidin; a favourable *milieu* to develop inflammation-related iron deficiency (Ganz, 2019). Accordingly, obesity is characterised by high levels of circulating inflammatory mediators, such as

CRP (Bernhardt et al., 2022), which is inversely correlated with serum iron levels in humans (Laudisio et al., 2023). Hepcidin has been recognised as the key molecular link between inflammatory diseases and iron handling (Ganz, 2003). Mechanistically, hepcidin is synthesised by hepatocytes and once released into the circulation, it binds and blocks ferroportin, an iron transporter found in many tissues, such as duodenal enterocytes, hepatocytes and reticuloendothelial macrophages. The sequestration of iron in these compartments reduces

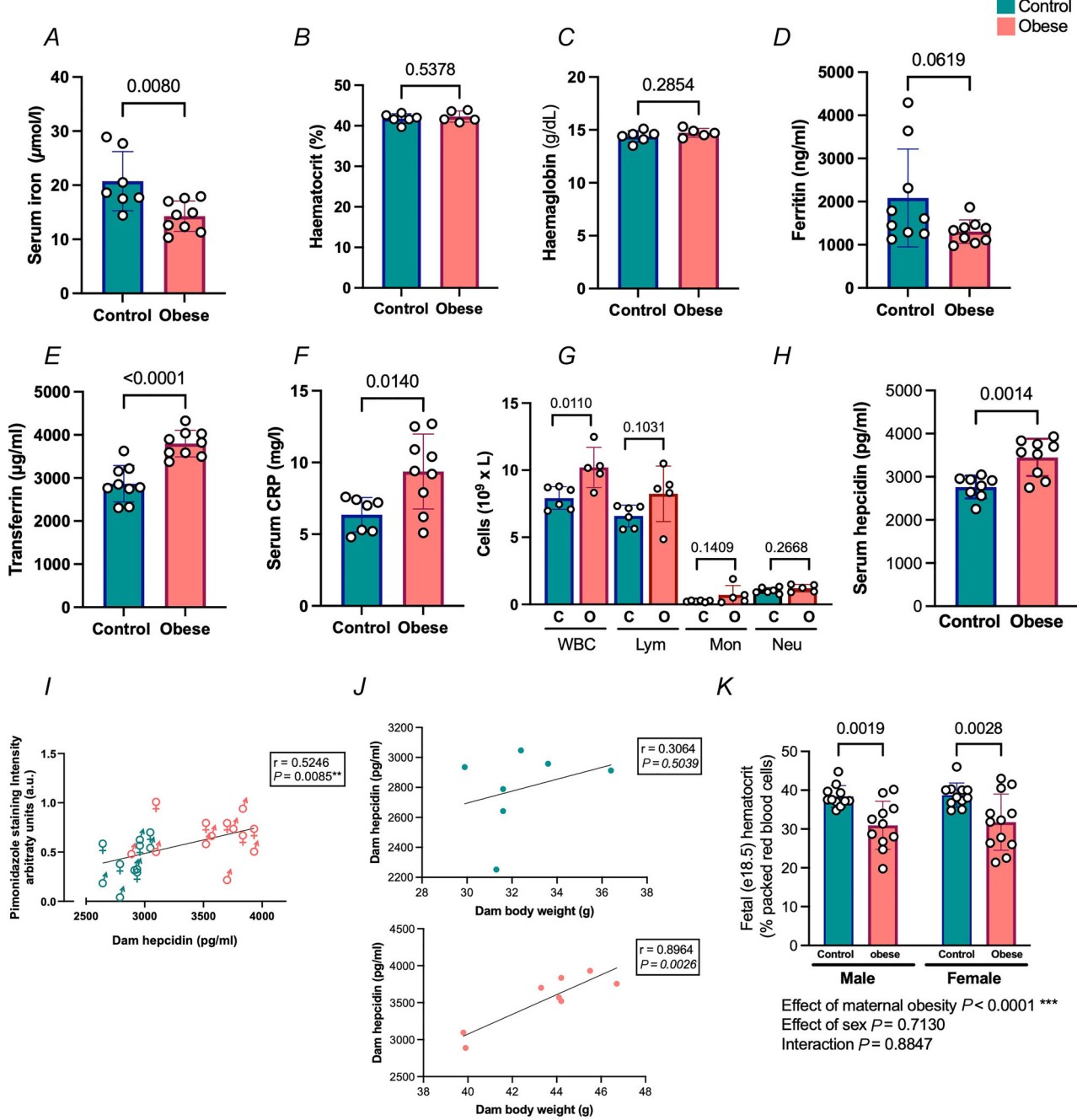

**Figure 5. Iron homeostasis and inflammation is altered by maternal obesity with concomitant correlation between maternal hepcidin and fetal hypoxia**

*A–F*, E13.5 maternal circulating iron levels (*A*), haematocrit (%) (*B*), haemoglobin (g/dl) (*C*), circulating ferritin levels (*D*), circulating transferrin levels (*E*), and circulating C-reactive protein (CRP) levels (*F*). *G*, white blood cells profile in E13.5 dams: white blood cells (WBC, $10^9$/l), lymphocytes (Lym, $10^9$/l), monocytes (Mon, $10^9$/l) and neutrophils (Neu, $10^9$/l). *H*, circulating hepcidin levels. *I*, Pearson's correlation between fetal whole torso anti-pimonidazole staining intensity and dam circulating hepcidin levels at E13.5. *J*, correlations between dam hepcidin levels in circulation and dam body weight at E13.5. Blue symbols: control dams; red symbols: obese dams. All correlations were assessed by the calculation of Pearson's correlation coefficient. *r* and *P*-values are indicated. *n* = 7–8 per condition. All data are presented as means ± SD and statistically assessed by Student's unpaired *t* test. *r* and *P*-values are indicated. *n* = 7–9 dams per condition. *K*, haematocrit in E18.5 offspring from control and obese dams. Differences were assessed by two-way ANOVA. *P*-values are indicated. *n* =11–12 animals per condition.

its bioavailability for iron-dependent processes (Nemeth & Ganz, 2021). Other inflammatory mediators, such as interleukin-1 and interleukin-6 can induce hepcidin levels (Lee et al., 2005; Nemeth et al., 2003, 2004) magnifying this response. Overnutrition is associated with increased risk of iron deficiency (Phillips et al., 2014; Tan et al., 2024), although there are also reports in the opposite direction, showing a direct correlation between haemoglobin levels and BMI in pregnant women (Sissala et al., 2022). Our results in mice support the idea that obesity disrupts iron homeostasis in the dam and establishes a positive association between the degree of hypoxia observed in fetuses of obese pregnancy with dam hepcidin levels, in concordance with previous reports in humans indicating a positive correlation between android fat and hepcidin levels (Stoffel et al., 2020). Different molecules induced by obesity can upregulate hepcidin levels. For instance, administration of leptin, a peptide hormone secreted by adipocytes and upregulated in obesity, in mice genetically deficient in leptin (*ob/ob* mice) acts on the liver to stimulate hepcidin production (Yamamoto et al., 2018). Consistent with this observation, leptin positively correlates with circulating hepcidin in humans (del Giudice et al., 2009). In addition, insulin, which is also increased by obesity, increases hepcidin mRNA and protein expression in a human hepatic cell line (HepG2 cells) (Wang et al., 2014). The positive correlation between serum leptin/insulin and hepcidin levels in obese mothers in the present study is also consistent with these findings.

Since iron levels were reduced in the maternal circulation, it is possible that fetal hypoxia induced by maternal obesity could result from poor maternofetal oxygen transfer through the placenta. This process relies heavily on the oxygen-carrying capacity of the maternal blood, which is in turn dependent on iron concentration in the maternal circulation. However, we do not favour this possibility as a fall in the maternal oxygen-carrying capacity would also lead to placental hypoxia, which we did not see in this study. Rather, in our mouse model of maternal obesity, it is more likely that fetal hypoxia occurred because of altered iron homeostasis, as the fetal iron balance relies on transferrin-bound iron uptake from the maternal circulation by placental cells and iron efflux from the placenta to the fetus. Obesity could affect iron transport through the placenta to the fetus via inflammatory pathways since its regulation is highly influenced by circulating hepcidin levels (Evans et al., 2011). In support of this hypothesis, in Jeg-3 choriocarcinoma cells, a model for human placental trophoblast, hepcidin induced a reduction in the expression of ferroportin and transferrin receptor (TfR), accompanied by reduced cellular export of iron. Further, high levels of maternal hepcidin were associated with lower expression of TfR on the syncytiotrophoblast (McDonald et al., 2022).

## Conclusions

Our findings suggest that obesity during pregnancy leads to maternal iron deficiency through multiple mechanisms including low-grade inflammation, elevated insulin, and leptin, that may independently or cooperatively increase the levels of the iron-regulating hormone hepcidin in the maternal circulation. We suggest this leads to reduced iron availability, resulting in fetal hypoxia, which is known to have causative effects on long-term cardiometabolic health (Giussani, 2021). These inflammation-induced defects in iron handling in obese pregnancies may make such pregnancies more susceptible to iron deficiency than lean pregnancies. Therefore, iron deficiency in mothers with obesity is a tractable therapeutic target for intervention that could prevent transmission of poor cardiometabolic health from mother to child. However, oral iron supplementation, may be less effective in obese pregnancy as the high hepcidin levels would bind to and induce ferroportin intracellular degradation in enterocytes, reducing maternal iron intestinal absorption. Therefore, alternative therapeutic strategies to prevent the detrimental effects of exposure to maternal obesity on the fetus are likely required, which could include the direct targeting of oxygen-sensing pathways in the fetus.

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

## Additional information

### Data availability statement

All analysed data can be found in the manuscript, raw datasets are available upon request.

### Competing interests

The authors declare they have no competing interests.

### Author contributions

A.C.-C.: investigation, analyses, writing and editing. I.I.: investigation, methodology, data acquisition and analyses, writing – original draft, review, and editing. A.H.: investigation, writing – review and editing. D.G.: investigation, analyses, writing – review and editing. D.S.F.-T.: conceptualization, funding acquisition, methodology, data acquisition and analyses, investigation, project administration, writing – review and editing. S.E.O.: project lead, conceptualization, visualization, funding acquisition, resources, supervision, writing – review and editing. All authors have read and approved the final version of this manuscript and agree to be accountable for all aspects of the work in ensuring that questions related to the accuracy or integrity of any part of the work are appropriately investigated and resolved. All persons designated as authors qualify for authorship, and all those who qualify for authorship are listed.

### Funding

This work was funded by British Heart Foundation (RG/17/12/33167) to S.E.O., D.S.F.-T. and D.G. and the Medical Research Council (MRC_MC_UU_00014/4) to S.E.O. and D.S.F-T. I. I. was a recipient of a British Heart Foundation PhD studentship (FS/18/56/35177) and A.C.-C. is the recipient of a Beca Postdoctorado en el Extranjero fellowship (ANID no. 74220049). A.H. was a recipient of a Wellcome Trust studentship (108926/B/15/Z).

### Acknowledgements

We thank Claire Custance and Tom Ashmore for their technical support.

### Keywords

developmental programming, fetal hypoxia, hepcidin, iron homeostasis, maternal obesity

### Supporting information

Additional supporting information can be found online in the Supporting Information section at the end of the HTML view of the article. Supporting information files available:

**Peer Review History**

