## [Peer Review History · The Journal of Physiology]

Maternal obesity during pregnancy disrupts iron homeostasis and promotes fetal hypoxia in the mouse

Adriana Cordova-Casanova, Isabella Inzani, Antonia Hufnagel, Dino A Giussani, Denise S Fernandez-Twinn, and Susan E Ozanne

DOI: 10.1113/JP288635

Corresponding author(s): *Adriana Cordova-Casanova (ap2427@cam.ac.uk)*

The following individual(s) involved in review of this submission have agreed to reveal their identity: *Kimberley J Botting-Lawford (Referee #2)*

Review Timeline:

Submission Date:	03-Feb-2025
Editorial Decision:	07-Apr-2025
Revision Received:	08-Aug-2025
Editorial Decision:	10-Sep-2025
Revision Received:	08-Oct-2025
Accepted:	27-Oct-2025

Senior Editor: *Laura Bennet*

Reviewing Editor: *Janna Morrison*

Transaction Report:

Dear Dr Cordova-Casanova,

Re: JP-RP-2025-288635 "**Maternal obesity during pregnancy disrupts iron homeostasis and promotes fetal hypoxia in the mouse**" by Adriana Cordova-Casanova, Isabella Inzani, Dino A Giussani, Denise S Fernandez-Twinn, and Susan E Ozanne

Thank you for submitting your manuscript to The Journal of Physiology. It has been assessed by a Reviewing Editor and by 2 expert referees and we are pleased to tell you that it is potentially acceptable for publication following satisfactory major revision.

LANGUAGE EDITING AND SUPPORT FOR PUBLICATION: If you would like help with English language editing, or other article preparation support, Wiley Editing Services offers expert help, including English Language Editing, as well as translation, manuscript formatting, and figure formatting at www.wileyauthors.com/eoo/preparation. You can also find resources for Preparing Your Article for general guidance about writing and preparing your manuscript at www.wileyauthors.com/eoo/prepresources.

REVISION CHECKLIST:

We look forward to receiving your revised submission.

Yours sincerely,

Laura Bennet
Senior Editor
The Journal of Physiology

REQUIRED ITEMS

- Author photo and profile. First or joint first authors are asked to provide a short biography (no more than 100 words for one author or 150 words in total for joint first authors) and a portrait photograph. These should be uploaded and clearly labelled together in a Word document with the revised version of the manuscript. See Information for Authors for further details.

- You must start the Methods section with a paragraph headed Ethical approval (https://jp.msubmit.net/cgi-bin/main.plex?form_type=display_requirements#methods).

Research must comply with The Journal's policies regarding animal experiments (<https://physoc.onlinelibrary.wiley.com/hub/animal-experiments>) and adherence to these policies must be stated in the manuscript.

Authors should confirm in their Methods section that their experiments were carried out according to the guidelines laid down by their institution's animal welfare committee, including an ethics approval reference number. The Methods section must contain a statement about access to food, water and housing, details of the anaesthetic regime: anaesthetic used, dose and route of administration, and method of killing the experimental animals.

- Please upload separate high-quality figure files via the submission form.

- Please ensure that the Article File you upload is a Word file.

- Papers must comply with the Statistics Policy: https://jp.msubmit.net/cgi-bin/main.plex?form_type=display_requirements#statistics.

In summary:

- If $n \leq 30$, all data points must be plotted in the figure in a way that reveals their range and distribution. A bar graph with data points overlaid, a box and whisker plot or a violin plot (preferably with data points included) are acceptable formats.

- If $n > 30$, then the entire raw dataset must be made available either as supporting information, or hosted on a not-for-profit repository, e.g. FigShare, with access details provided in the manuscript.

- 'n' clearly defined (e.g. x cells from y slices in z animals) in the Methods. Authors should be mindful of pseudoreplication.

- All relevant 'n' values must be clearly stated in the main text, figures and tables.

- The most appropriate summary statistic (e.g. mean or median and standard deviation) must be used. Standard Error of the Mean (SEM) alone is not permitted.

- Exact p values must be stated. Authors must not use 'greater than' or 'less than'. Exact p values must be stated to three significant figures even when 'no statistical significance' is claimed.

EDITOR COMMENTS

Reviewing Editor: Methods Details:
Please add details as per reviewers comments.

Comments to the Author:

This is an interesting paper about iron homeostasis. This is not a well studied area during pregnancy.

The analysis for the effects of obesity and sex is a strength. However, it would be most helpful to show the results of the 2 way ANOVA as the main effects and the interaction.

Figure 2 does not report volume but rather area. Is it possible that on the section measured there is no change in area but if multiple sections were measured there would be a change in area? Or is this data meant to represent degrees of hypoxia? Is this possible? My understanding is that this probe activated when tissue $PO_2 < 10\text{mmHg}$ and is more an on-off indicator rather than a direct measure of PO_2 .

For some regressions, it may be relevant to perform the regression in the normal and obese groups separately and then determine if the slopes are different or if relationships only exist on one group or the other.

REFEREE COMMENTS

Referee #1:

The manuscript by Cordova-Casanova et al presents an interesting study. The authors investigated a potential link between maternal obesity and fetal hypoxia. The exploration of inflammation induced iron deficiency adds a curious layer to the study. However, there do not appear to be any strong relationships or correlations between inflammatory markers and markers of iron deficiency. Whilst correlations do not indicate causation, it would be expected that should the author's obesity-inflammation-iron deficiency be correct- some relationships between these markers would exist. Aside from this, the manuscript is well written; although, I would urge the authors not to be too discussive in the results section- save this for the discussion. See below my specific comments that I hope the authors can clearly address:

1. Line 40: the Bertossa reference is not a mouse model, it is in fact a NHP model- which likely offers more translatable power than mouse models. The sentence reads like this reference is being grouped in with mouse models- please correct.
2. Line 132: what post hoc test was used to isolate differences between groups if the interaction between maternal diet and fetal sex was significant?
3. Line 92- more information on the TD MRI is required. ie sequence parameters.
4. Was anaesthesia required for the TD-MRI- if so, this should be clearly stated.
5. The method of humane killing- were there differences in the time taken for obese vs control dams to die? Did obese dams require more or less increments of CO_2 inhalation? Could this have accounted for the increased pimonidazole abundance?
6. Is there data available that shows the relationship between pimonidazole abundance/intensity and the partial pressure of oxygen in tissue? Understanding what level of tissue hypoxia is being induced by maternal obesity would be an important factor for further studies in larger animal models.
7. The authors have previously used a sheep model of maternal obesity- do they care to comment on whether iron deficiency was induced in that larger animal model?
8. The study has a large focus on maternal anaemia. The authors have not provided specific iron intake for the dams. What was the iron composition of the control vs the obesogenic diet? Could they have become anaemic due to lower iron in the obese chow rather than the hypothesized inflammatory pathway?

9. Figure 3- suggest either define DAB in legend or change y axis to pimonidazole staining intensity. Provide units.
10. From Figure 5F- there are two "control" dams with elevated hepcidin? Was there anything different about these dams? Was an outliers test performed- ie Grubbs? I note this because it appears they are driving the relationship with your marker of fetal torso hypoxia and not any of the those dams in the obese group.
11. Have the authors looked at any relationships between obese markers (fat mass etc) and the dams actual inflammatory markers? That would give you insight as to whether the "obesity" was driving maternal inflammation. Perhaps a multivariate correlation is in order here. Fat mass to inflammatory marker- to iron deficiency - to fetal hypoxia.
12. Line 189: "These results indicate that increased fetal hypoxia in obese pregnancies compared with controls is not due to impaired placental vascularisation at e13.5 / 0.7 of gestation or increased placental hypoxia." - This is the type of sentence that should be in the discussion- not the results section.
13. Line 230: "The lack of a fetal brain sparing effect triggered by fetal hypoxia at e13.5 / 0.7 of gestation is likely due to immaturity of the chemoreflex, endocrine and local redox mechanisms that initiate and maintain this fetal cardiovascular defence" - prior to this statement the authors reference their previous work at a later gestational age- did they observe "brain sparing" in that study when the mechanisms behind this physiology may be more mature?
14. Line 241: "Prenatal iron deficiency (3 mg iron / kg diet)" - what was the amount of iron given the control diet of the study this sentence goes on to reference? Perhaps between to say the % decrease in iron from the control group.
15. Table 2: both groups are titled "control dams"
16. Supplementary figure 1- the post hoc p values are listed, so is the main factor p value for the effect of maternal obesity. Where is the p value for the effect of fetal sex ?
17. Supp figure 3- this shows important data linking iron homeostasis to the maternal obesogenic phenotype- not sure why this is being hidden away as a supp figure.
18. The obese dams had significantly lower relative heart weight- can the authors comment on this? In the human setting - would an increase in relative heart weight be more likely as a result of cardiac hypertrophy?
19. What about maternal haemoglobin or hematocrit levels? Were they different between groups?

Referee #2:

The current manuscript presented by Cordova-Casanova and colleagues explores a possible link between maternal obesity, iron homeostasis and fetal hypoxia. This is an important topic, but I have some concerns that I would liked addressed.

I feel the evidence of a causative link between maternal iron homeostasis and fetal hypoxia is currently lacking. I also do not see any evidence that the fetus from obese pregnancy has altered iron homeostasis themselves, which you suggest is the cause for the fetal hypoxia. A few extra pieces of data will strengthen this work. As it stands, I believe the conclusions are too definitive and need editing, for example, Line 220 'In the current study, we establish that inflammation and disruption of iron homeostasis in obesogenic pregnancies is linked to fetal hypoxia'.

I assume there wasn't enough fetal serum to measure fetal iron/ferritin/transferrin. It would have been possible, though, to collect enough blood to determine fetal Hct as a proxy measure of impaired red blood cell production. Please provide additional evidence if possible.

It is possible that the increase in maternal transferrin is compensatory and the fetus is unaffected by the low maternal iron/ferritin? Has placental ferroportin protein abundance been affected by maternal obesity?

I see you have used a mouse secondary antibody for your pimonidazole immuno. Please provide negative control image/data for immuno when the primary antibody has been omitted. This negative control will demonstrate there has not

been any non-specific binding of your secondary antibody to endogenous mouse immunoglobins. This is critical to your project and hypothesis, since inflammation can increase immunoglobins. This immuno is the only evidence that you have that the fetus is hypoxic, so this negative control will make your data more compelling.

I have some concerns regarding your correlations when there is a known significant difference between your groups for the X and Y variables, especially for Mean DAB intensity vs Dam fat mass. On visual inspection it looks as though these apparently strong correlations are simply because the control and obese group means differ. Please also calculate correlations within each group separately to see if there is a true relationship between fetal hypoxia and maternal measures within each group. Alternatively, apply a regression analysis where you control for the group effect.

I cannot find any detail on the anesthetic protocol or recovery for the weekly TD-MNR scans.

Minor:

Graphical abstract, find alternative wording for 'increased hypoxia degree' that improves clarity.

Line 25, WHO reference is from 2000, please replace with something more current.

Table 3, title for obese group mislabelled as 'Control dams'.

END OF COMMENTS

We thank the editor and reviewers for the very helpful and constructive comments that have helped improve the manuscript. We have addressed them all (including addition of new data) and provide a point-by-point response to them below.

EDITOR COMMENTS

Comments to the Author:

This is an interesting paper about iron homeostasis. This is not a well-studied area during pregnancy.

We agree with the editor that iron homeostasis is an underexplored area during pregnancy.

The analysis for the effects of obesity and sex is a strength. However, it would be most helpful to show the results of the 2-way ANOVA as the main effects and the interaction.

We agree with the reviewer that the effects of obesity and sex make the paper strength. In the revised version, we have included the results of the 2-way ANOVA with the main effects and the interaction in all our figures showing results in the offspring. Please see figures 2, 4, 5K and supplementary figure 1.

Figure 2 does not report volume but rather area. Is it possible that on the section measured there is no change in area but if multiple sections were measured there would be a change in area? Or is this data meant to represent degrees of hypoxia? Is this possible? My understanding is that this probe activated when tissue PO₂<10mmHg and is more an on-off indicator rather than a direct measure of PO₂.

As the reviewer highlighted, pimonidazole is a 2-nitroimidazole compound expected to be activated when tissue PO₂ is under 10mmHg (PMID: 7759162). Therefore, pimonidazole staining positively correlates with pO₂ less than 10mmHg (PMID: 10319731).

Our results indicate basal activation of the probe in fetal tissues and placenta in control animals (as has been reported previously PMID: 31002746, PMID: 24801305). Therefore, we considered that the most appropriate comparison would be based on the intensity of the stain in comparable sections of tissues (PMID: 25968580, PMID: 25368228, PMID: 24401554) from fetuses of control and obese dams. We did not measure area of the stain to avoid the issue mentioned by the editor regarding the potential differences in the stained areas depending on depth level where the section was done. To clarify our approach, we have now explicitly stated in line 133 that our quantification is based on staining intensity.

For some regressions, it may be relevant to perform the regression in the normal and obese groups separately and then determine if the slopes are different or if relationships only exist on one group or the other.

We thank the reviewer for raising this important point. Please see in the table below the joint and separate correlations for control and obese groups presented in figure 3 and 5. In most cases there is no significant correlation when the groups are studied separately (presumably due to lack of power) and therefore the need to combine data from both groups. For clarity we have used different colours to indicate the data points originating from the different maternal groups. The only exception to this was the correlation between dam hepcidin to dam body weight where a significant difference was only observed in the obese group. We have therefore included a correlation graph for these two parameters for each maternal group separately. Please see new supplementary figure 3.

Correlations fig 3	r	p value
Fetal hypoxia vs dam body weight	0.4902	0.0110*
Fetal hypoxia vs dam body weight (control)	0.3738	0.2083
Fetal hypoxia vs dam body weight (obese)	0.0667	0.8284

Fetal hypoxia vs dam fat mass	0.4309	0.0280*
Fetal hypoxia vs dam fat mass (control)	-0.0640	0.8337
Fetal hypoxia vs dam fat mass (obese)	-0.0576	0.8515
Fetal hypoxia vs dam insulin	0.3936	0.0467*
Fetal hypoxia vs dam insulin (control)	-0.1133	0.7124
Fetal hypoxia vs dam insulin (obese)	0.3924	0.1847
Correlations fig 5		
Fetal hypoxia vs dam hepcidin	0.4861	0.0118*
Fetal hypoxia vs dam hepcidin (control)	0.4911	0.1250
Fetal hypoxia vs dam hepcidin (obese)	0.0437	0.8873
Dam hepcidin vs dam body weight	0.8799	<0.0001****
Dam hepcidin vs dam body weight (control)	0.3064	0.5039
Dam hepcidin vs dam body weight (obese)	0.8964	0.0026*
Dam hepcidin vs fat mass	0.8462	<0.0001****
Dam hepcidin vs fat mass (control)	0.7427	0.0558
Dam hepcidin vs fat mass (obese)	0.6435	0.0851
Dam hepcidin vs leptin	0.8114	0.0004***
Dam hepcidin vs leptin (control)	-0.2203	0.6351
Dam hepcidin vs leptin (obese)	0.3454	0.4480
Dam hepcidin vs insulin	0.6297	0.0119*
Dam hepcidin vs insulin (control)	0.4913	0.2628
Dam hepcidin vs insulin (obese)	0.4001	0.3260

REFEREE

Referee

COMMENTS

#1:

The manuscript by Cordova-Casanova et al presents an interesting study. The authors investigated a potential link between maternal obesity and fetal hypoxia. The exploration of inflammation induced iron deficiency adds a curious layer to the study. However, there do not appear to be any strong relationships or correlations between inflammatory markers and markers of iron deficiency. Whilst correlations do not indicate causation, it would be expected that should the author's obesity-inflammation-iron deficiency be correct- some relationships between these markers would exist. Aside from this, the manuscript is well written; although, I would urge the authors not to be too discussive in the results section- save this for the discussion.

We thank the reviewer was raising this comment. To address this, we have performed new correlations, and the results were included in a new table (Table 4) and discussed accordingly in the text (lines 223 - 224). We have also modified the manuscript discussion as recommended by the referee.

See below my specific comments that I hope the authors can clearly address:
1. Line 40: the Bertossa reference is not a mouse model, it is in fact a NHP model- which likely offers more translatable power than mouse models. The sentence reads like this reference is being grouped in with mouse models- please correct.

We thank the reviewer for the comment. This reference has been moved from the introduction to the discussion with the species clearly indicated (please see line 230 - 234).

2. Line132: what post hoc test was used to isolate differences between groups if the interaction between maternal diet and fetal sex was significant?

We did not find any statistically significant interaction between maternal diet and fetal sex. We used Tukey's as a post hoc test to isolate differences between groups where appropriate and this information is now incorporated into the statistics section of the methods (line 141-142).

3. Line 92- more information on the TD MRI is required. ie sequence parameters.

4. Was anaesthesia required for the TD-MRI- if so, this should be clearly stated.

We have expanded the information regarding our TD-NMR protocol. Anaesthesia was not required; animals are just temporarily restrained; we hope the text is clear now (Please see lines 96 - 101).

5. The method of humane killing- were there differences in the time taken for obese vs control dams to die? Did obese dams require more or less increments of CO2 inhalation? Could this have accounted for the increased pimonidazole abundance?

We have no have evidence to suggest that there are differences in the CO2 inhalation between groups or time for animals to die. Both experimental groups were exposed to the same raising flow of CO2 for 3 minutes.

6. Is there data available that shows the relationship between pimonidazole abundance/intensity and the partial pressure of oxygen in tissue? Understanding what level of tissue hypoxia is being induced by maternal obesity would be an important factor for further studies in larger animal models.

We do not have our own data regarding the relationship between pimonidazole and partial pressure of oxygen in tissue. We used this staining approach based on previous reports indicating that pimonidazole is expected to be activated when tissue PO2 is under 10mmHg (PMID: 7759162) and a positive correlation between PO2 and pimonidazole staining has been reported previously (PMID: 10319731). We agree that having data from larger animals would be very positive to compare with our findings in mice and hope that other researchers with expertise in large animal models take that forward in the future.

7. The authors have previously used a sheep model of maternal obesity- do they care to comment on whether iron deficiency was induced in that larger animal model?

Unfortunately, we have no data in relation to iron deficiency in the sheep model of maternal diet-induced obesity.

8. The study has a large focus on maternal anaemia. The authors have not provided specific iron intake for the dams. What was the iron composition of the control vs the obesogenic diet? Could they have become anaemic due to lower iron in the obese chow rather than the hypothesized inflammatory pathway?

We thank the reviewer for this important comment. In our study serum iron is reduced in the obese dams (Figure 5A) without reaching anaemia, measured as haemoglobin or haematocrit levels please see new figure supplementary 4. We believe that the reduced iron levels in the obese dams are triggered by low degree inflammation for two reasons:

1) The dietary iron in the diet of both groups, control and obese are not iron deficient and are over the minimal required for mice pregnancy in mice (PMID: 23582419). The iron composition in both diets is now included in the manuscript (lines 83 – 85).

2) Low iron levels in the circulation triggered by a reduced dietary iron intake are associated with reduced levels of hepcidin (PMID: 20932599, PMID: 23589789). In contrast, low iron levels in circulation caused by low degree inflammation are expected to be associated with high levels of hepcidin. In our animal model of maternal obesity the low iron occurs in the presence of high levels of inflammatory markers (CRP) and hepcidin suggesting that it is the latter of the two causes of low iron occurring.

9. Figure 3- suggest either define DAB in legend or change y axis to pimonidazole staining intensity. Provide units.

We have changed the y axis labelling to pimonidazole staining intensity which is expressed in arbitrary units. (please see figures 2, 4C and supplementary figure 1 C, D and E).

10. From Figure 5F- there are two "control" dams with elevated hepcidin? Was there anything different about these dams? Was an outliers test performed- ie Grubbs? I note this because it appears they are driving the relationship with your marker of fetal torso hypoxia and not any of the those dams in the obese group.

We detected a mistake in the correlation graph (figure 5F) as the values of hepcidin plotted correspond to the hepcidin levels divided by 20 (the working dilution for the ELISA test) and not to the cleared data. We apologies for this mistake. Two outliers were detected via a ROUT test and have now been excluded (we extended the outliers detection to all out datasets). We performed a new correlation test excluding outlier values and the result was even more statistically significant with p values of 0.0118* vs 0.0085**. Please see the graphs below. The corrected version of the graph has now been included in the paper (Figure 5F).

11. Have the authors looked at any relationships between obese markers (fat mass etc) and the dams actual inflammatory markers? That would give you insight as to whether the "obesity" was driving maternal inflammation. Perhaps a multivariate correlation is in order here. Fat mass to inflammatory marker- to iron deficiency - to fetal hypoxia.

In our model, maternal obesity-related indicators such as body weight, fat mass and insulin correlate with fetal hypoxia (Figure 3A-C). In addition, these same maternal obesity-related indicators correlated with maternal hepcidin levels (Figure 5G-J) and hepcidin correlates with fetal hypoxia (Figure 5F).

To complement our results, we performed new correlation tests between factors in the inflammation / low iron / fetal hypoxia axis. We thank the reviewer for the suggestion. We have incorporated this new information in table 4.

Table 4

Correlation	r	p value
Dam fat mass vs IL6	0.5652	0.0026**
Dam fat mass vs CRP	0.5527	0.0023**
Dam fat mass vs dam hepcidin	0.8461	<0.0001****
Dam fat mass vs dam ferritin	-0.4835	0.0051**
Dam fat mass vs fetal hypoxia	0.3894	0.0493*
Dam hepcidin vs dam iron	-0.4563	0.0376*

Dam hepcidin vs fetal hypoxia	0.4700	0.0205*
Dam hepcidin vs dam ferritin	-0.4706	0.0087**
Dam hepcidin vs dam transferrin	0.7181	<0.0001****
Dam ferritin vs fetal hypoxia	-0.3949	0.0459*
Dam Iron vs dam transferrin (ug/ml)	-0.6258	0.0014**
Dam transferrin vs CRP	0.7222	<0.0001****

12. Line 189: "These results indicate that increased fetal hypoxia in obese pregnancies compared with controls is not due to impaired placental vascularisation at E13.5 / 0.7 of gestation or increased placental hypoxia." - This is the type of sentence that should be in the discussion- not the results section.

We thank the reviewer for the comment. This conclusion has now been moved to the discussion (lines 241-246).

13. Line 230: "The lack of a fetal brain sparing effect triggered by fetal hypoxia at E13.5 / 0.7 of gestation is likely due to immaturity of the chemoreflex, endocrine and local redox mechanisms that initiate and maintain this fetal cardiovascular defence" - prior to this statement the authors reference their previous work at a later gestational age- did they observe "brain sparing" in that study when the mechanisms behind this physiology may be more mature?

Our previous publication (PMID: 34505282) shows there was not a brain sparing effect in the E18.5 fetuses of obese dams. However, hypoxia was not explored as a readout in that work to directly compare the results of E13.5 and E18.5 fetuses.

14. Line 241: "Prenatal iron deficiency (3 mg iron / kg diet)" - what was the amount of iron given the control diet of the study this sentence goes on to reference? Perhaps between to say the % decrease in iron from the control group.

In the cited study, the amount of iron given to the control group was 37 mg iron / kg of diet. Therefore, the iron deficient diet they used contained only 8.1% of the control diet. We have now included this information in the manuscript (line 258 - 259).

15. Table 2: both groups are titled "control dams"

We apologise for the typographical error which has been corrected.

16. Supplementary figure 1- the post hoc p values are listed, so is the main factor p value for the effect of maternal obesity. Where is the p value for the effect of fetal sex?

We have now included the p values for fetal sex, and interaction in all the figures presenting offspring data. (please see figures 2, 4, 5K, and supplementary 1)

17. Supp figure 3- this shows important data linking iron homeostasis to the maternal obesogenic phenotype- not sure why this is being hidden away as a supp figure.

We agree that hepcidin and maternal obesity phenotype correlation is important data. Following the reviewer's suggestion this is now presented in figure 5G, H, I, J.

18. The obese dams had significantly lower relative heart weight- can the authors comment on this? In the human setting - would an increase in relative heart weight be more likely as a result of cardiac hypertrophy?

We report that the ratio heart / body weight is reduced in obese dams. This is what would be expected as the dams have such a large increase in fat mass. The obese dams in our study have at least 10 grams of fat (compared with ~ 5 grams in control dams) therefore is expected

that the relative mass of lean tissue such as the heart in relation to body weight would be reduced. To detect cardiac hypertrophy, other analysis would need to be performed, such as heart weight / tibia length or heart weight / lean mass. This could be followed by cellular and molecular analysis (PMID: 25051449) but unfortunately, we don't have such data for the dam hearts. We have incorporated the ratio of the organ weights / lean mass. Please see table 1.

19. What about maternal haemoglobin or haematocrit levels? Were they different between groups?

To address this question, we have performed a haematological profile of the dams at E13.5 of pregnancy.

Maternal haemoglobin or haematocrit levels were not affected by obesity, meaning that the reduction in iron levels were not enough to trigger anaemia in the mother at E13.5.

Our results indicate that there is a statistically significant increase in the absolute number of white blood cells. Which could be interpreted as another sign of systemic inflammation. Please see the graphs below for more details. This data has now been included in the manuscript. Please see line 204, 212 / supplementary figure 4.

Referee

#2:

The current manuscript presented by Cordova-Casanova and colleagues explores a possible link between maternal obesity, iron homeostasis and fetal hypoxia. This is an important topic, but I have some concerns that I would liked addressed.

1. I feel the evidence of a causative link between maternal iron homeostasis and fetal hypoxia is currently lacking. I also do not see any evidence that the fetus from obese pregnancy has altered iron homeostasis themselves, which you suggest is the cause for the fetal hypoxia. A few extra pieces of data will strengthen this work. As it stands, I believe the conclusions are too definitive and need editing, for example, Line 220 'In the current study, we establish that inflammation and disruption of iron homeostasis

in obesogenic pregnancies is linked to fetal hypoxia'.

We thank the reviewer for their comment. We have modified the conclusions in both the Discussion and the Abstract to prevent any potential misunderstanding. Our study characterizes maternal and fetal phenotypes, establishing correlations that suggest a potential mechanism and we agree with the reviewer that this cannot establish causality.

We agree that strengthening the data related to fetal iron status is important. It was not possible to directly measure serum iron levels in E13.5 fetuses (due to technical limitations of blood volume available). We have measured the fetal haematocrit at E18.5 where such analysis is feasible and observed a statistically significant reduction, which is likely indicative of decreased iron levels at this later stage of gestation. We have incorporated this new figure in the revised version of the manuscript lines 224 - 227 (Figure 5K).

2. I assume there wasn't enough fetal serum to measure fetal iron/ferritin/transferrin. It would have been possible, though, to collect enough blood to determine fetal Hct as a proxy measure of impaired red blood cell production. Please provide additional evidence if possible.

It is possible that the increase in maternal transferrin is compensatory, and the fetus is unaffected by the low maternal iron/ferritin? Has placental ferroportin protein abundance been affected by maternal obesity?

Unfortunately, it is not technically feasible to get enough sample from E13.5 mouse fetuses to study fetal iron/ferritin/transferrin/haematocrit levels. However, we were able to measure the haematocrit in the offspring of control and obese dams at E18.5 (Figure 5K). Our results align with our hypothesis indicating that fetuses from obese dams have reduced haematocrit compared with fetuses from control dams at E18.5. This also demonstrate that the increased transferrin levels in the obese dams are not enough to compensate for the low iron levels and the fetuses are affected, at least at this later stage of gestation, as is evidenced by the reduction in the haematocrit.

Regarding the ferroportin expression in the placenta, RNA sequencing data at E18.5 indicates that there are no differences in ferroportin (Slc40a1) mRNA levels between the two groups. We also showed that the expression of other iron homeostasis-related genes in the placenta was not affected by maternal obesity. Please see the table below.

Ensembl gene ID	Gene name	Log(FC)	p-value	FDR
ENSMUSG00000025993	Slc40a1	-0.19799	0.153719	0.988285
ENSMUSG00000022797	Tfrc	-0.71519	0.001803	0.296718
ENSMUSG00000024661	Fth1	-0.00406	0.978814	0.999957
ENSMUSG00000026389	Steap3	0.266848	0.196172	0.999957
ENSMUSG00000012428	Steap4	0.51297	0.293099	0.999957
ENSMUSG00000053897	Slc39a8	-0.0202	0.958769	0.999957
ENSMUSG00000022094	Slc39a14	-0.23965	0.017936	0.62134
ENSMUSG00000023030	Slc11a2	0.033355	0.828348	0.999957
ENSMUSG00000051695	Pcbp1	0.125142	0.282917	0.999957
ENSMUSG00000056851	Pcbp2	0.105514	0.292855	0.999957
ENSMUSG00000003617	Cp	-0.4117	0.010963	0.526204
ENSMUSG00000031209	Heph	0.157144	0.778075	0.999957
ENSMUSG00000022032	Scara5	-0.25553	0.081195	0.89103
ENSMUSG00000040249	Lrp1	0.211093	0.081056	0.89103

3. I see you have used a mouse secondary antibody for your pimonidazole immuno. Please provide negative control image/data for immuno when the primary antibody has been omitted. This negative control will demonstrate there has not been any non-specific binding of your secondary antibody to endogenous mouse immunoglobins.

This is critical to your project and hypothesis, since inflammation can increase immunoglobins. This immuno is the only evidence that you have that the fetus is hypoxic, so this negative control will make your data more compelling.

We thank the reviewer for this comment that we are happy to address. We have indeed done such a negative control and did not detect any evidence for an increase in non-specific signal in the obese group. Please see the images below.

20C2 (control) – negative stain control.

20F32 (obese) - negative stain control

4. I have some concerns regarding your correlations when there is a known significant difference between your groups for the X and Y variables, especially for Mean DAB intensity vs Dam fat mass. On visual inspection it looks as though these apparently strong correlations are simply because the control and obese group means differ. Please also calculate correlations within each group separately to see if there is a true relationship between fetal hypoxia and maternal measures within each group. Alternatively, apply a regression analysis where you control for the group effect.

Thank you for this comment. Please find a detailed response to this comment alongside a table with the separate correlations for control and obese groups in the response to the editor section above.

5. I cannot find any detail on the anesthetic protocol or recovery for the weekly TD-MNR scans.

Anaesthesia was not required for the TD-MNR measurements. We have added this information to the methods (line 96 – 101).

Minor:

6. Graphical abstract, find alternative wording for 'increased hypoxia degree' that improves clarity.

We had changed increased hypoxia degree to increased relative hypoxia levels for clarity, considering that all our results are being compared to control situation.

7. Line 25, WHO reference is from 2000, please replace with something more current.
We thank for the observation. We have replaced the reference with a newer one.

8. Table 3, title for obese group mislabelled as 'Control dams'.
We apologise for the mistake. The title of table 3 has been now corrected.

Dear Dr Cordova-Casanova,

Re: JP-RP-2025-288635R1 "**Maternal obesity during pregnancy disrupts iron homeostasis and promotes fetal hypoxia in the mouse**" by Adriana Cordova-Casanova, Isabella Inzani, Antonia Hufnagel, Dino A Giussani, Denise S Fernandez-Twinn, and Susan E Ozanne

Thank you for submitting your revised Research Article to The Journal of Physiology. It has been assessed by the original Reviewing Editor and Referees and has been well received. Some final revisions have been requested.

REVISION CHECKLIST:

Please upload two versions of your manuscript text: one with all relevant changes highlighted and one clean version with no changes tracked. The manuscript file should include all tables and figure legends, but each figure/graph should be uploaded as separate, high-resolution files. The journal is now integrated with Wiley's Image Checking service. For further details, see: <https://www.wiley.com/en-us/network/publishing/research-publishing/trending-stories/upholding-image-integrity-wileys->

image-screening-service

We look forward to receiving your revised submission.

Yours sincerely,

Laura Bennet
Senior Editor
The Journal of Physiology

REQUIRED ITEMS

- Your paper contains Supporting Information of a type that we no longer publish, including supplementary tables and figures. Any information essential to an understanding of the paper must be included as part of the main manuscript and figures. The only Supporting Information that we publish are video and audio, 3D structures, program codes and large data files. Your revised paper will be returned to you if it does not adhere to our Supporting Information Guidelines.

EDITOR COMMENTS

Reviewing Editor:

Thank you for revising the paper. JP does not allow supplementary data. As per the reviewer's request, please include supplementary data in the paper.

REFEREE COMMENTS

Referee #1:

Thank you for adequately addressing my comments.

Referee #2:

Thank you to the authors for their response. The additional data provided strengthens the study and alleviates my concerns. Where possible, I would recommend providing the data presented in the response to reviewers in the supplementary materials to satisfy any skeptical readers. Please also ensure the methods for any supplementary data is also provided if it is not included in the primary manuscript. For example, I couldn't find the methods for maternal haematology.

END OF COMMENTS

We thank the editor and reviewers for their comments, which we believe have significantly improved the manuscript. We provide a point-by-point response to them below.

Reviewing Editor:

Thank you for revising the paper. JP does not allow supplementary data. As per the reviewer's request, please include supplementary data in the paper.

Thank you. We have incorporated all the supplementary data in the paper.

Referee #1:

Thank you for adequately addressing my comments.

We thank the referee #1 for their helpful suggestions and were happy to make their suggested edits.

Referee #2:

Thank you to the authors for their response. The additional data provided strengthens the study and alleviates my concerns. Where possible, I would recommend providing the data presented in the response to reviewers in the supplementary materials to satisfy any skeptical readers. Please also ensure the methods for any supplementary data are also provided if it is not included in the primary manuscript. For example, I couldn't find the methods for maternal haematology.

We thank the referee #2 for their comments.

We thank Referee #2 for their comments and suggestions for improving the paper. We have incorporated the new data (presented to reviewers) and supplementary figures into the paper. We also added the missing methods information regarding maternal haematology.

Dear Dr Cordova-Casanova,

Re: JP-RP-2025-288635R2 "**Maternal obesity during pregnancy disrupts iron homeostasis and promotes fetal hypoxia in the mouse**" by Adriana Cordova-Casanova, Isabella Inzani, Antonia Hufnagel, Dino A Giussani, Denise S Fernandez-Twinn, and Susan E Ozanne

We are pleased to tell you that your paper has been accepted for publication in The Journal of Physiology.

Yours sincerely,

Laura Bennet
Senior Editor
The Journal of Physiology

IMPORTANT POINTS TO NOTE FOLLOWING ACCEPTANCE OF YOUR PAPER:

- You can help your research get the attention it deserves! Check out Wiley's free Promotion Guide for best-practice recommendations for promoting your work at: www.wileyauthors.com/eoo/guide. You can learn more about Wiley Editing Services which offers professional video, design, and writing services to create shareable video abstracts, infographics, conference posters, lay summaries, and research news stories for your research at: www.wileyauthors.com/eoo/promotion.

- If you would like to receive our 'Research Roundup', a monthly newsletter highlighting the cutting-edge research published in The Physiological Society's family of journals (The Journal of Physiology, Experimental Physiology, Physiological Reports, The Journal of Nutritional Physiology and The Journal of Precision Medicine: Health and Disease), please click this link, fill in your name and email address and select 'Research Roundup': <https://www.physoc.org/journals-and-media/membernews>

EDITOR COMMENTS

Reviewing Editor:

Thank you for revising the paper.

REFeree COMMENTS

Referee #2:

Thank you to the authors for including the additional data. I am satisfied with the response to all points raised.